# Corrosion Behavior of High-Strength C71500 Copper-Nickel Alloy in Simulated Seawater with High Concentration of Sulfide

**DOI:** 10.3390/ma15238513

**Published:** 2022-11-29

**Authors:** Xin Gao, Ming Liu

**Affiliations:** 1Research and Development Department, Beijing Med-Zenith Medical Scientific Corporation Limited, Beijing 101316, China; 2Key Laboratory of Biomechanics and Mechanobiology, Ministry of Education, Beijing Advanced Innovation Center for Biomedical Engineering, School of Biological Science and Medical Engineering, Beihang University, Beijing 100083, China; 3Center for Advancing Materials Performance from the Nanoscale (CAMP-Nano), State Key Laboratory for Mechanical Behavior of Materials, Xi’an Jiaotong University, Xi’an 710049, China

**Keywords:** C71500 copper-nickel alloy, sulfide, immersion test, corrosion behavior, seawater environment

## Abstract

The corrosion behavior of high-strength C71500 copper-nickel alloy in high concentrations of sulfide-polluted seawater was studied by potentiodynamic polarization measurements, electrochemical impedance spectroscopy (EIS), immersion testing, and combined with SEM, EDS, XPS, and XRD surface analysis methods. The results showed that the C71500 alloy shows activation polarization during the entire corrosion process, the corrosion rate is much higher (0.15 mm/a) at the initial stage of immersion, and the appearance of diffusion limitation by corrosion product formation was in line with the appearance of a Warburg element in the EIS fitting after 24 h of immersion. As the corrosion process progressed, the formed dark-brown corrosion product film had a certain protective effect preventing the alloy from corrosion, and the corrosion rate gradually decreased. After 168 h of immersion, the corrosion rate stabilized at about 0.09 mm/a. The alloy was uniformly corroded, and the corrosion products were mainly composed of Cu_2_S, CuS, Cu_2_(OH)_3_Cl, Mn_2_O_3_, Mn_2_O, MnS_2_, FeO(OH), etc. The content of Cu_2_S gradually increased with the extension of immersion time. The addition of S^2−^ caused a large amount of dissolution of Fe and Ni, and prevented the simultaneous formation of a more protective Cu_2_O film, which promoted the corrosion process to some extent.

## 1. Introduction

Copper-nickel alloys are widely used in marine environments because of their good corrosion resistance, machinability, high thermal conductivity, and electrical conductivity, as well as their moderate biological scaling resistance. They can be utilized in many marine engineering structures, such as seawater desalination, seawater reverse irrigation generators, marine ship power systems, marine ship power generation systems, etc. [1,2]. The corrosion rate of copper-nickel alloy pipes in polluted seawater is much faster than that in unpolluted areas; corrosion resistance can be lost in the presence of sulfide ions and other sulfur-containing substances [3,4]. The sulfide in polluted seawater in coastal areas can originate from industrial waste discharge, biological and bacterial reproduction processes in seawater (algae, marine organisms or microorganisms, bacteria that reduce sulfide), and atmospheric sulfide emissions in coastal areas, which may all also cause seawater pollution [5,6].

Copper and its alloys have been widely studied in seawater environments. Kong et al. [7] studied the effect of sulfide concentrations on copper corrosion behavior in anaerobic chloride solution, and found that the corrosion resistance of copper decreased with the increase of sulfide concentration and sulfide addition. Chen et al. [8] demonstrated that the migration of Cu occurred between the surface of the net anode and cathode by using a pre-corroded and fresh Cu electrode in NaCl solution containing sulfide. Rao and Kumar [6,9] studied the corrosion inhibition behavior of copper-nickel alloy in simulated seawater and synthetic seawater containing 10 ppm sulfide, which confirmed that 5-(3-Aminophenyl) tetrazole shows good corrosion inhibition. Radovanović et al. [10] studied the protective effect of the non-toxic compound 2-amino- 5-ethyl- 1,3,4-thiadiazole (AETDA) on copper in acidic chloride solution, and the results showed that the stability of the protective layer mainly depends on the concentration of the inhibitor and the potential value of the protective film. Nady et al. [11] studied the electrochemical properties of Cu and Cu-10Al-10Ni in sulfide ion containing 3.5 wt.% NaCl solution, and found that the addition of Ni into the Cu_2_O barrier film could enhance the corrosion resistance of the alloy. Jandaghi et al. [12] conducted in-depth research on the microstructure evolution and corrosion resistance of aluminum/copper joints manufactured by explosive welding processes and revealed the cause of corrosion.

Our group [13,14,15,16,17] has previously studied the grain boundary engineering treatment on the mechanical behavior of C71500 copper-nickel alloy. Using the process of “multi pass deformation and grain boundary treatment + single pass deformation recovery” could obtain a large number of low coincidence site lattice grain boundary structures. In order to improve the corrosion resistance of the alloy, the harmful elements such as C, S, and O are reduced to less than 5 ppm by vacuum refining technology; the strength of the material reaches 430 MPa, and the elongation rate reaches 45%. Additionally, we found that this alloy has good corrosion resistance in a 3.5 wt.% NaCl simulated seawater environment [18]. In the actual use process, it is found that the copper-nickel alloy cooling equipment stably runs during the process of navigation, but after the ship berths for a long time, the corrosion damage of cooling equipment often occurs. It is found that the S^2−^ content of flowing seawater is not high, but in the non-flowing area of the pipeline, the S^2−^ content of seawater reaches 0.5 wt.%; the corrosion behavior is obviously different from previous research results [19,20]. Under such conditions, stress corrosion cracking easily occurs. In the marine environment, a large number of sulfur ions exist due to the discharge of industrial wastes, the reproduction process of organisms and bacteria in seawater, and the discharge of atmospheric sulfide in coastal areas. The corrosion behavior of seawater polluted by sulfur ions is significantly different from previous studies [21]. Based on this, the corrosion behavior of the C71500 Cu-Ni alloy in high concentration S^2−^ polluted seawater was systematically studied.

## 2. Experimental Procedure

### 2.1. Specimen and Solution

The material used in this study was C71500 copper-nickel alloy manufactured according to ASTM-B224. The chemical composition (wt.%) was: Ni = 30.54, Mn = 0.93, and Fe = 0.80, the content of C, S, and O was strictly controlled, and the residual was Cu. After casting, forging, hot piercing, cold rolling and heat treatment, the material was fabricated into a pipe with the dimensions of Ø 60 × 5.03 mm. In order to ensure the original structure of the material, all samples were directly cut from the alloy pipe. The as-revived microstructure of the alloy was a single-phase, and a large number of low coincidence site lattice (CSL) grain boundaries could be obtained after thermomechanical treatment [13,14,15,16,17].

The sample for the immersion test was cut into four equal parts in the radial direction, at a length of 35 mm. The electrochemical test specimen was cut with a dimension of 10 × 10 mm toward the pipe wall thickness; the outer surface of the pipe was used as the testing surface, and the remaining part was sealed with epoxy resin. The testing arc surface of the pipe was ground by silicon carbide paper down to 1200#. The outer diameter, wall thickness, and length of the sample were measured by a micrometer and the surface area of the sample was drawn using SolidWorks software. Then, the sample was ultrasonically degreased in acetone for 10 min, and put into a dish to dry for 24 h before testing.

A solution of 3.5% NaCl + 0.5% Na_2_S (wt.%) was chosen to simulate a high concentration sulfide-containing seawater environment with a pH 8.20. All experiments were carried out in a constant temperature water bath at around 35 ± 2 °C.

### 2.2. Immersion Test

The immersion test was performed according to ASTM G31-2012a standard. The exposure time was set to 6 durations of 24, 72, 120, 168, 336, and 672 h. Five samples were used for each duration; three were used for weight loss testing, and the other two were used for surface analysis. After each period, the corrosion products of weight loss samples were removed by HCl solution with a volume ratio (HCl/H_2_O) of 2: 1. The weight of samples before and after immersion testing were measured by an electronic balance (METTLER TOLEDO ME204/02) with an accuracy of 0.0001 g. The corrosion rate can be obtained by Equation (1):(1)$$v=\frac{8{.76(\;m}_{0}{\;-m}_{1})}{S_{0}\mathrm{td}}$$
where: *v* is the corrosion rate, mm/a; *m*_0_ is the mass of the sample before corrosion, g; *m*_1_ is the mass of the sample after removing corrosion products, g; *S*_0_ is the area of the tested surface, m^2^; *t* is the corrosion test cycle, h; and *d* is the density of the alloy, g/cm^3^.

### 2.3. Electrochemical Measurement

A CS2350H/CORRTEST electrochemical workstation with a three electrode system was used to carry out the electrochemical analysis, the working electrode was an arc sample, the saturated calomel electrode (SCE) was used as a reference electrode and a platinum mesh electrode with a surface area of 8 cm^2^ was the auxiliary electrode. The open circuit potential (OCP) was tested for 30 min before other electrochemical features were measured. The frequency range (100 kHz–0.01 Hz) of electrochemical impedance spectrum (EIS) was selected and the AC amplitude signal was 10 mV. Potentiodynamic polarization was performed from cathodic 0.3 V (OCP) to anodic 0.8 V (vs. OCP) with a scan rate of 1 mV/s.

In order to study the electrochemical corrosion characteristics of the alloy at different immersion times, we also tested the samples immersed in the corrosion solution for different periods, as for the immersion test procedure.

### 2.4. Surface Analysis

A Zeiss Gemini 300 scanning electron microscope (SEM) was applied for surface observation, and the relevant corrosion product was analyzed by EDS.

The corroded product compositions were analyzed by X-ray diffractometer (Rigaku SmartLab 9 kW) in the range of 20°–100° and a step length of 0.02°, and Jade 6.5 software was used to analyze the test results.

The corrosion products were quantificationally analyzed by XPS diffractometer (Thermo Fisher Scientific Escalab 250Xi). The scanning range, interval, and pitch was 0–1200 eV, 1 eV, and 0.1 eV, respectively, with a spot diameter of 400 μm. Peak fitting and analysis was performed using Advantage software and the binding energy calibration standard was the C1s spectral line (284.8 eV).

## 3. Results and Discussion

### 3.1. Weight Loss Experiment

Figure 1 shows the C71500 alloy average corrosion rate variation with different corrosion times. As can be seen from Figure 1, the maximum corrosion rate of the alloy is 0.15 mm/a at the initial stage of immersion. With the extension of immersion time, the average corrosion rate of the alloy rapidly decreases. When the corrosion time reaches 168 h, the average corrosion rate of the C71500 alloy tends to be stable at about 0.09 mm/a. This change may be related to the formation and densification of the protective corrosion product film on the surface of the alloy. With the formation and dissolution of the protective film gradually balanced, the corrosion rate of the alloy after long-term immersion tends to be stable.

### 3.2. Potentiodynamic Polarization Measurements

Figure 2 shows the potentiodynamic polarization curves of the alloy under different corrosion conditions. It can be seen from Figure 2a that the electrochemical response of C71500 alloy in 3.5 wt.% NaCl solution is significantly different by adding S^2−^. The corrosion potential of C71500 alloy shifts to the negative direction and the anodic curve is no longer smooth. It can be seen from Figure 2b that the polarization curves of C71500 alloy in simulated polluted marine solution at immersion for various times shift to the left, i.e., lower current densities, and the corrosion rate decreases with immersion time. It should be noted that one or more current density decrease inflection points could be observed before the peak current density is reached after adding 0.5 wt.% Na_2_S. With increasing immersion time, the anodic polarization curves of the alloy change: at the early stage of corrosion, two secondary anodic peaks appear at about −525 mV and −404 mV (SCE) [5,22]. When the immersion time increases to 336 h and 672 h, the peak of −525 mV (SCE) disappears, and only a −404 mV (SCE) peak can be observed. The Tafel extrapolation method was applied to fit the parameters of the polarization curves at various immersion times. It can be seen from Table 1 that, at the initial stage of immersion, the corrosion current density (*i*_corr_) is much higher, which is about 96.6 μA/cm^2^; *i*_corr_ decreases and tends to stabilize for longer immersion times, which is consistent with the corrosion weight loss results presented in Figure 1.

### 3.3. Electrochemical Impedance Spectroscopy

The Nyquist and Bode plots of C71500 alloy immersed in simulated polluted marine solution for different periods are shown in Figure 3. It can be seen from the Nyquist diagram that only a small capacitive arc could be observed after immersion for 30 min, which indicates that the corrosion rate is much higher at the initial stage of immersion [23]. Warburg impedance could be seen in the low frequency parts of the Nyquist plots after immersion for 24 and 72 h, which indicates that the speed of corrosion process is faster and the substances involved in the reaction are quickly consumed, and that the characteristic of the diffusion control process may exist [4]; after 120 h, the diffusion impedance disappears and the capacitive reactance arc radius further increases, which indicates the corrosion rate is greatly reduced. The modulus of impedance |Z| in Figure 3b in the low frequency region gradually increases with exposure time and tends to be stable at immersion for 120 h. Therefore, the corrosion process can be mainly divided into three stages: the first stage (0–72 h) includes the formation of a corrosion product layer; the second stage (72–120 h) is the transition stage; the third stage (120–672 h) is the stable corrosion product film. The appearance of Warburg impedance reflects the growth process of the corrosion product film and its related mass transfer limitations. With the increase in immersion time, the arc radius of the capacitive gradually increases (72–672 h), indicating increasing corrosion resistance, and then tends to stabilize at even longer immersion times, which indicates that the formed corrosion product film is in a dynamic equilibrium process of growth/dissolution, thus reducing the corrosion rate. 

Based on prior research [24,25,26], the equivalent circuits (Figure 4) were used to fit the EIS of C71500 alloy after immersion in simulated polluted marine solution for various times; where *R*_s_
*R*_t_, *R*_f_, CPE_1_, CPE_2_, and *W* is the solution resistance, charge transfer resistance, corrosion product film resistance, constant phase element of the corrosion product film (Q_dl_), the constant phase element (Q_f_) related to charge transfer, and finite length diffusion Warburg impedance, *Z*_W_ [27] respectively. The Q_dl_ impedance can be obtained by [27]:(2)$$\frac{1}{Z}=Q_{\mathrm{dl}}{\left(j\omega \right)}^{n}$$
where j, ω, and *n* are the imaginary number (j^2^ = −1), the angular frequency (ω = 2 *pf*) and the exponent of Q_dl_, respectively. When *n* = 1, it relates to resistance; *n* = −1 relates to an inductance; and *n* = 0.5 is equal to Warburg impedance [28]. The CPE is often used as a pure capacitance to eliminate the dispersion effect caused by electrode surface roughness and for other reasons in an analog equivalent circuit [18].

The equivalent circuit parameters fitted by ZsimpWin software are shown in Table 2. With increasing immersion time, *R*_t_ gradually increases and reaches the maximum value of 6619 Ω·cm^2^ after immersion for 672 h, indicating the corrosion rate gradually decreases with immersion time. Q_dl_ also increases with immersion time, which indicates the gradual growth of a corrosion product film. *R*_f_ reaches the maximum value at about 120 h, then decreases and increases again, which indicates the dynamic equilibrium of the formed corrosion product film. Q_f_ is the double layer capacitance associated with corrosion products, which indicates that the corrosion product film may be changed with immersion times.

### 3.4. Morphology Analysis

#### 3.4.1. Macroscopic Morphology

Figure 5 shows the macro morphology of C71500 alloy after immersion in simulated polluted marine solution for various times. It can be seen that the alloy mainly shows uniform corrosion. After immersion for 24 h, the sample surface is covered with a dark brown corrosion products layer, and the corrosion degree gradually increases with immersion time. The corrosion products fall off and deposit at the bottom of the solution with immersion time, thus resulting in different corrosion morphologies at local areas of the sample.

#### 3.4.2. SEM Analysis

In order to further analyze the evolution of corrosion products on C71500 alloy at high lateral resolution, the corrosion product morphology after different immersion times were analyzed by SEM. In the initial 24 h of immersion (Figure 6a), less than 1 μm granular products are formed on the surface, and the gap between particles is larger. The particle size gradually increases with immersion time, and flat sheet corrosion product films could be locally detected, the surface of the specimen is not completely covered by the corrosion product film, and the film is in its lateral growth stage [29]. After immersion for 72 h, it can be seen that the surface of the alloy is completely covered by the granular corrosion product layer, and the size of the particles gradually increases from less than 1 to 3 μm; there are also some scattered corrosion films between the corrosion products. The particles continue to increase when the immersion time reaches 120 h, and the thickness of the corrosion product film also further increases. Small spheres are difficult to form by precipitation from supersaturated solutions [30]. After 168 h of immersion, cracks can be seen in the corrosion product film, and the cracks show grain boundary related patterns, which indicates that there may be local intergranular corrosion in C71500 alloy; the initial corrosion product film also shows as irregular (see Figure 6m). After 336 h of immersion, a large amount of corrosion products adhere to the surface of the sample, and the formed film is loose and porous. After immersion for 672 h, the formed corrosion product film has gradually become dense. It can be seen that the corrosion morphology of the alloy has greatly changed with various immersion times. The SEM morphologies shown in Figure 6d–i) indicate the shape of spherical corrosion particles at the beginning of immersion, and the edges and corners become more and more clear with immersion time.

The corrosion product EDS analysis of C71500 alloy after different immersion times is shown in Table 3, and the surface scan energy spectrum analysis is depicted in Figure 7. It can be seen that the content of Fe and Ni in the corrosion products decreases with immersion time, which indicates that the alloy is subject to Fe and Ni removal [1]; the content of Cu in the corrosion products increases. The content of S fluctuates; it obviously increases in the early stage, indicating that S may participate in the corrosion reaction and could be incorporated in the corrosion product on the substrate surface.

### 3.5. XPS Analysis

The corrosion product film composition of samples after immersion for 72, 168, and 336 h was analyzed by XPS (Figure 8). It can be seen that the corrosion product film is mainly composed of Cu, S, Cl, Mn, O, and Na. After 72 h of immersion, a small amount of Fe-based corrosion products are detected on the surface of the sample, while it disappears after longer immersion times, and a small amount of Ni-based corrosion products are seen after 336 h of immersion.

#### 3.5.1. Cu Spectrum

In Table 2, the p peak for all tested samples are composed of the Cu 2p1/2 (952 eV) and Cu 2p3/2 (932.2 eV) peaks (Figure 9a). Except for a slight fluctuation in the early stage of immersion (72 h), no obvious S polarization and satellite peaks were detected. Since Cu and Cu_2_O compounds do not have the excitation peak of the Cu 2p3/2 spectrum, it can be preliminarily concluded that the superficial corrosion product films do not contain Cu^2+^. It shows that Cu is mainly composed of Cu^+^, except for a small amount of Cu^2+^ that may be formed in the early stages of immersion. The Auger spectrum of Cu was analyzed to further verify the valence state of Cu (Figure 9b). It shows that the Cu Auger spectrum is between 916.4 and 916.8 eV, again establishing that the corrosion product is composed of Cu^+^ [31,32].

The Cu 2p3/2 spectra of the corrosion product after various immersion times are shown in Figure 10. The binding spectrum was analyzed in high-resolution conditions (929–939 eV). The corrosion products are mainly comprised CuS, Cu_2_S, and Cu_2_O. The binding energy peak of Cu was 160,000 CPS in the early stages of immersion, indicating more Cu could be observed in the corrosion film. After immersion for 168 h, the binding energy peak of Cu 2p3/2 stabilized at 75,000–85,000 CPS. Only Cu_2_S and CuS could be detected in the 72-h immersed sample, indicating that a protective Cu_2_O layer had not been formed. The Cu_2_O film was considered to be a uniform and dense corrosion product film that adhered well to the substrate and promoted passivity of the copper-nickel alloy, thereby improving the corrosion resistance of the alloy [33]. When the immersion time reached 168 h, Cu_2_O could be observed. In this work, the CuS component gradually decreased and the Cu_2_O component gradually stabilized with immersion time (see Table 4). 

#### 3.5.2. Mn Spectrum

The binding spectrum of Mn 3s under high-resolution conditions (72.5–97.5 eV) is shown in Figure 11a. No small satellite peaks could be detected (72 h), while the energy differences between the two satellite peaks (168 and 336 h) were 6.0 and 6.1 eV, respectively, indicating that the Mn^2+^ was the main product in the corrosion product film.

The Mn 2p spectra after different immersion times are shown in Figure 11b–d. It can be seen that there is no obvious difference between the peak values. The composition of the corrosion product after 72-h immersion is mainly Mn, MnO, and MnCl_2_, among which the content of Mn is the least. With the extension of the immersion time, MnS and high Mn-oxides (MnO_2_, Mn_2_O_3_) are observed. For prolonged immersion times, the high Mn-oxide products disappear and the components become MnS, MnO, and MnCl_2_. The Ni, Mn, Fe, and other alloying elements are oxidized into high-oxides at the metal/film interface, and then migrate at the corrosion product membrane/solution interface. High-price ions increase the ionic and electronic resistance of the Cu_2_O, thereby improving the corrosion resistance of Cu alloys [34]. This also means that corrosion-resistant Cu alloys can be improved by forming high-oxide ions (see Table 5).

#### 3.5.3. O Spectrum

The O 1s spectrum of the C71500 alloy corrosion product after different immersion times is shown in Figure 12. The oxide after the initial immersion is mainly composed of Cu_2_O. With increasing immersion time, Ni_2_O_3_ is observed in the corrosion product film. Ni atoms could be oxidized in the form of NiO or Ni_2_O_3_, and doped in the Cu_2_O film interacting with O_2_ (see Table 6).

#### 3.5.4. S Spectrum

The corrosion product layer S 2p spectra after different immersion times are shown in Figure 13. The composition of the corrosion products is not stable at the early stages of immersion, the peak spectrum of the S 2p after 72 h immersion is as low as 6400 CPS, and many saw-tooth patterns are observed on the waveform. The corrosion products are mainly Cu_2_S, FeS_2_, CuS, and some of the interference peaks of Na_2_S and Na(SO_3_)_2_. When the immersion time reaches 168 h, the waveform of the S 2p spectrum smoothens; FeS components are detected at an immersion time of 336 h (see Table 7).

### 3.6. XRD Analysis

Figure 14 shows the XRD pattern of the C71500 alloy surface corrosion product layer after different immersion times. The observed corrosion products are mainly Cu_2_(OH)_3_Cl, CuS, Cu_2_S, Mn_2_O_3_, Mn_2_O, MnS_2_, and FeO(OH). Among them, the peak distribution of Cu_2_S is the most obvious, and increases with immersion time. After 72 h of immersion, the corrosion product is mainly Cu_2_(OH)_3_Cl and FeO(OH); the peak of Cu_2_(OH)_3_Cl gradually disappears and a large amount of Cu_2_S is detected at an immersion time of 168h, the accompanying peaks of Mn_2_O, Mn_2_O_3_, and MnS_2_ could also be observed; the accompanying peaks of Cu_2_(OH)_3_Cl and Cu_2_S in the corrosion products significantly increase when the immersion time reaches 336 h.

## 4. Corrosion Mechanism 

### 4.1. Cu Reaction Mechanism

Cu is ionized through electron transfer with Cl^−^, resulting in ${\mathrm{CuCl}}_{\mathrm{ads}}$ [35].
(3)$${\mathrm{Cu}+\mathrm{Cl}}^{-}\to {\mathrm{CuCl}}_{\mathrm{ads}\;}{+e}^{-}$$

It is easy to dissolve and form ${\mathrm{CuCl}}_{2}^{-}$ [27,35]:(4)$${\mathrm{CuCl}}_{\mathrm{ads}\;}{+\mathrm{Cl}}^{-}\to {\mathrm{CuCl}}_{2}^{-}$$

In the neutral solution, the cathode reaction can be determined as follows:(5)$$O_{2}\;{+2H}_{2}{O+4e}^{-}\to {4\mathrm{OH}}^{-}$$

The C71500 alloy polarization behavior in chloride containing solution mainly dissolves Cu in soluble ${\mathrm{CuCl}}_{2}^{-}$ according to the process described above [36,37]. In the case of high chloride concentration, ${\mathrm{CuCl}}_{3}^{2-}$ and ${\mathrm{CuCl}}_{4}^{3-}$ [35,38] may also be formed. Subsequently, in the environment of ${\mathrm{OH}}^{-}$, Cu_2_O is generated through the following reaction:(6)$${2\mathrm{CuCl}}_{2}^{-}{+2\mathrm{OH}}^{-}\to {\mathrm{Cu}}_{2}{O+H}_{2}{O+4\mathrm{Cl}}^{-}$$

It is also possible to hydrolyze the resulting soluble ion complex to form a corrosion-resistant Cu_2_O layer [36].
(7)$${2\mathrm{CuCl}}_{2}^{-}{+H}_{2}O\to {\mathrm{Cu}}_{2}{O+4\mathrm{Cl}}^{-}\;{+2H}^{+}$$

Under dissolved oxygen conditions, as immersion time increases, the corrosion product is CuO [39]. Stable Cu_2_O can be oxidized to CuO through the following series of reactions [40]:(8)$${\mathrm{Cu}}_{2}{O+O}_{2}{+H}_{2}O\to {2\mathrm{CuO}+H}_{2}O_{2}$$
(9)$${\mathrm{Cu}}_{2}{O+H}_{2}O_{2}\to 2{\mathrm{CuO}+H}_{2}O$$

In the XRD analysis, ${\mathrm{Cu}}_{2}{\left(\mathrm{OH}\right)}_{3}\mathrm{Cl}$ could be detected, and then ${\mathrm{Cu}(\mathrm{OH})}_{2}$ and ${\mathrm{Cu}}_{2}{\left(\mathrm{OH}\right)}_{3}\mathrm{Cl}$ are formed by the reaction of Cu_2_O and H_2_O [40]:(10)$${\mathrm{Cu}}_{2}{O+3H}_{2}O\to {2\mathrm{Cu}(\mathrm{OH})}_{2}{+2H}^{+}{+2e}^{-}$$
(11)$${\mathrm{Cu}}_{2}{O+2H}_{2}{O+\mathrm{Cl}}^{-}\to {\mathrm{Cu}}_{2}{\left(\mathrm{OH}\right)}_{3}{\mathrm{Cl}+H}^{+}$$

In addition, local acidification of corrosion products could induce CuO to dissolve and generate ${\mathrm{Cu}}_{2}{\left(\mathrm{OH}\right)}_{3}\mathrm{Cl}$ [41]:(12)$${2\mathrm{CuO}+\mathrm{Cl}}^{-}{+H}^{+}{+H}_{2}O\to {\mathrm{Cu}}_{2}{\left(\mathrm{OH}\right)}_{3}\mathrm{Cl}$$

The corrosion behavior of C71500 alloy is related to the stability of the formed corrosion product layer. When exposed to an aerated water solution, the copper surface undergoes an electrochemical transformation to form an oxide film. A double ${\mathrm{CuCl}}_{\mathrm{ads}\;}$/${\mathrm{Cu}}_{2}O$ layer will be formed in the initial stage, and then a pure Cu_2_O layer will eventually be formed [42]. The protectiveness of the film will be increased with an extension of immersion time, and can also be significantly improved due to the presence of Ni and Fe elements [37].

The above analysis is the first immersion corrosion reaction stage of the C71500 alloy, which is the same as the corrosion process without adding Na_2_S. The specific reaction process is shown in Figure 15a.

After adding sulfides, sulfur can be hydrogenated to produce ${\mathrm{HS}}^{-}$ [43]:(13)$$S^{2-}{+H}_{2}O↔{\mathrm{HS}}^{-}{+\mathrm{OH}}^{-}$$

As can be seen from the anodic polarization curves (Figure 2b), a secondary anode peak could be observed. In all studied electrolytes, the current increases as the potential reaches a definite limit value, which remains unchanged within a large potential range of several hundreds of millivolts. The presence of sulfide ions increases the current limit, which may be caused by the deterioration of the protective film or the anodic oxidation of sulfide ions [44]:(14)$${\mathrm{HS}}^{-}{}_{\mathrm{aq}}↔S_{s}+H^{+}+2e^{-}$$
(15)$${3\mathrm{HS}}^{-}{}_{\mathrm{aq}}\to {\;S}_{3}^{2-}{+3H}^{+}{+4e}^{-}$$
(16)$${2\mathrm{HS}}^{-}{}_{\mathrm{aq}}{+2H}_{2}O\to S_{2}O_{3}{}^{2-}{+8H}^{+}\;{+8e}^{-}$$

Since a thin porous and non-protective Cu_2_S film is formed on the alloy surface, the corrosion rate and corrosion current will increase, thereby catalyzing the corrosion reaction and preventing the formation of a protective oxide layer [1,31]. The formation of Cu_2_S interferes with the protective oxide film and reduces its corrosion resistance [5]. The higher corrosion rate is due to the highly defective Cu_2_O layer containing Cu_2_S, which could lead to the rapid exchange of ions and electrons; Cu_2_S is not as protective as Cu_2_O [45].

The presence of sulfide ions increased the corrosion process of C71500 alloys in the studied electrolytes. The increase in corrosion rate is due to the formed film (porous Cu_2_S) which catalyzes the corrosion reaction and prevents the formation of a more protective corrosion product layer [46]. The presence of sulfide ions could promote the corrosion of Cu to form adsorbed sulfides. These adsorbed species catalyze the anode dissolution reaction through the following conditions [46]:(17)$${\mathrm{Cu}}_{s}{+\mathrm{HS}}^{-}\to {\mathrm{Cu}(\mathrm{HS}}^{-}{)}_{\mathrm{ads}}$$

Then, Cu may undergo anodic dissolution:(18)$${\mathrm{Cu}(\mathrm{HS}}^{-})\to \mathrm{Cu}{\left(\mathrm{HS}\right)}_{s}{+e}^{-}$$

Followed by the dissociation and reorganization process:(19)$$\mathrm{Cu}{\left(\mathrm{HS}\right)}_{s}\to {\mathrm{Cu}}^{+}{}_{\mathrm{aq}}{+\mathrm{HS}}^{-}{}_{\mathrm{aq}}$$
(20)$${2\mathrm{Cu}}^{+}+{\mathrm{HS}}^{-}{}_{\mathrm{aq}}+{\mathrm{OH}}^{-}{}_{\mathrm{aq}}\to {\mathrm{Cu}}_{2}S_{s}{+H}_{2}O$$

The overall reaction equation can be written as: (21)$${4\mathrm{Cu}+2S}^{2-}+O_{2}{+2H}_{2}O↔{2\mathrm{Cu}}_{2}{S+4\mathrm{OH}}^{-}$$

The formation of hydroxide is proceeded through the following reaction:(22)$${\mathrm{Cu}}^{+}{}_{\mathrm{aq}}+{\mathrm{OH}}^{-}{}_{\mathrm{aq}}\to \mathrm{Cu}{\left(\mathrm{OH}\right)}_{s}$$

In parallel with these reactions, Equation (7) could also form Cu_2_O. However, the presence of Cu_2_S greatly hinders the protectiveness of the Cu_2_O film. Therefore, the corrosion resistance of the film can be greatly reduced by increasing the concentration of sulfide ion.

Under low HS^−^ and high Cl^−^ concentration conditions, soluble chloride complexes ${\mathrm{CuCl}}_{2}^{-}$ formed by reacting with surface intermediates ${\mathrm{Cu}(\mathrm{HS}}^{-}{)}_{\mathrm{ads}}$ can compete with the film formation reaction [23]. This will lead to a large amount of Cu dissolution. When the HS^−^ is low on Cu alloy surface, Cu^+^ could be transferred to the Cu_2_S/solution interface by dissolving ${\mathrm{CuCl}}_{2}^{-}$ (Equation (23)), where the concentration of HS^−^ is much higher, and Cu_2_S would be formed by Equation (24) which will dominate at the film/electrolyte interface:(23)$${\mathrm{Cu}(\mathrm{HS}}^{-}{)}_{\mathrm{ads}}{+2\mathrm{Cl}}^{-}\to {\mathrm{CuCl}}_{2}^{-}{+\mathrm{HS}}^{-}$$
(24)$${2\mathrm{CuCl}}_{2}^{-}{+\mathrm{HS}}^{-}\to {\mathrm{Cu}}_{2}{S+4\mathrm{Cl}}^{-}{+H}^{+}$$

As can be seen from XPS results of samples at the early stages of immersion, the corrosion products in the outer layer are mainly Cu_2_S, Cu_2_O, and CuS, while the intermediate layer analyzed by XRD is mainly ${\mathrm{Cu}}_{2}{\left(\mathrm{OH}\right)}_{3}\mathrm{Cl}$. From the analysis of the spectra of O 1s and S 2p in XPS, it can be seen that the content of Cu_2_O is basically unchanged and the content of Cu_2_S gradually increases as the reaction progresses, indicating that Equation (24) is the main form of reaction at this time. A schematic diagram of the whole reaction process is shown in Figure 15b. 

### 4.2. Fe Reaction Mechanism

In the early stages of corrosion product formation, the dissolution (oxidation) kinetics of iron in seawater is much faster than that of copper [1]:(25)$$\mathrm{Fe}\to {\mathrm{Fe}}^{2+}\;{+2e}^{-}$$

Only a small part of the dissolved Cu can precipitate as Cu_2_O using Equation (6), and most of them dissolve in the solution due to hydrodynamic conditions. In contrast, most of the Fe-oxides could convert to γ-FeOOH through the following reactions [1]:(26)$${\mathrm{Fe}}^{2+}+2{\mathrm{OH}}^{-}\to \mathrm{Fe}(\mathrm{OH}{)}_{2}$$
(27)$${\mathrm{Fe}(\mathrm{OH})}_{2}\to {\mathrm{FeO}+H}_{2}O$$
(28)$$\mathrm{FeO}+H_{2}O\to \gamma -\mathrm{FeOOH}+H^{+}+e^{-}$$

The total formation reaction of γ-FeOOH in the copper-nickel alloy corrosion product film in seawater can be described as:(29)$$\mathrm{Fe}+2H_{2}O\to \gamma -\mathrm{FeOOH}+3H^{+}+3e^{-}$$

Therefore, although Fe is a trace element in the alloy, it can gradually accumulate and enrich in the form of γ-FeOOH in the corrosion product film. The content of γ-FeOOH in the corrosion product film can be used as an indicator to reflect the corrosion rate of copper-nickel alloys in seawater, because iron is oxidized and unevenly precipitated during the corrosion (Cu dissolution) process.

However, the function of γ-FeOOH in the corrosion product film of copper-nickel alloy is still controversial. Campbell et al. [41] reported that γ-FeOOH simultaneously precipitated with Cu_2_O at the early stage of corrosion to form a complete protective film, and γ-FeOOH also has an important effect on the cathode polarization reaction due to its high resistivity. Zanoni et al. [47] pointed out that the surface film of C71640 copper-nickel alloy is mainly composed of γ-FeOOH in seawater, the corrosion product does not adhere or protect under such low *E*_corr_, resulting in the active corrosion state of the alloy. Vreeland et al. [48] attributed the inhibitory effect of ferrous sulfate (FeSO_4_) on copper-nickel pipes to the deposition of a brown γ-FeOOH layer.

In the presence of a large amount of Na_2_S, NaCl and H_2_O first react to form acidic substances [49]:(30)$${2\mathrm{NaCl}+H}_{2}O\to \mathrm{NaOH}+2\mathrm{HCl}$$

Then, Fe reacts with acid:(31)$$2\mathrm{HCl}+\mathrm{Fe}\to {\mathrm{FeCl}}_{2}{+H}_{2}$$
(32)$${\mathrm{FeCl}}_{2}{+H}_{2}S\to \mathrm{FeS}+2\mathrm{HCl}$$

The total formation reaction of FeS in copper-nickel alloy in polluted seawater can be described as [45,46]:(33)$${\mathrm{Fe}+H}_{2}S\to {\mathrm{FeS}+H}_{2}$$

From the analysis of XPS, it can be seen that Equation (29) is the main reaction in the initial stage of immersion, and the corrosion product is mainly composed of γ-FeOOH. The content of H_2_S in the solution gradually increases with the extension of corrosion time, and thus Equation (33) becomes the main corrosion process; significant amounts of FeS could be observed, and the high-priced FeS_2_ also could be detected. The dissolution rate of Fe increases with the progress of the reaction, leading to a gradual decrease of Fe content in the EDS analysis. The specific schematic diagram of Fe dissolution process is illustrated in Figure 15b.

### 4.3. Ni Reaction Mechanism

The enriched Ni elements in the outer and the middle layer of the film are mainly in metallic form, and a small amount of Ni^2+^ could incorporate into ${\mathrm{Cu}}_{2}{\left(\mathrm{OH}\right)}_{3}\mathrm{Cl}$/Cu_2_O and replace Cu^2+^/Cu^+^ [50]. The corrosion rate of Ni is approximately two orders of magnitude lower than that of Cu [51]. The content of Ni incorporated in the Cu_2_O lattice accordingly increases when the metallic Ni is enriched at the alloy and film interface. The amorphous or microcrystalline NiO are transited by Cu_2_O in certain local areas, where the doped Ni concentration reaches a certain value, and then, the NiO phase grows through appropriate conditions. The production of Ni(OH)_2_ may be due to the NiO hydrolysis reaction caused by the dissolution of the outer layer and the penetration of seawater at certain locations with the increase of immersion time, this reaction process is explained in the formation process of the NiO inner surface layer in Figure 15c.

The presence of sulfide ions promotes the corrosion of nickel in a way similar to that of copper, and forms adsorbed sulfides on the alloy surface. These adsorbed species catalyze the anode dissolution reaction according to the following conditions [45]:(34)$${\mathrm{Ni}}_{s}{+\mathrm{HS}}^{-}\to {\mathrm{Ni}(\mathrm{HS}}^{-}{)}_{\mathrm{ads}}$$

Ni may undergo anodic dissolution:(35)$${\mathrm{Ni}(\mathrm{HS}}^{-})\to {\mathrm{Ni}(\mathrm{HS})}_{s}{+e}^{-}$$

Followed by the dissociation and reorganization process:(36)$${\mathrm{Ni}(\mathrm{HS})}_{s}\to {\mathrm{Ni}}^{+}{}_{\mathrm{aq}}{+\mathrm{HS}}^{-}{}_{\mathrm{aq}}$$
(37)$${2\mathrm{Ni}}^{+}{+\mathrm{HS}}^{-}{}_{\mathrm{aq}}{+\mathrm{OH}}^{-}{}_{\mathrm{aq}}\to {\mathrm{Ni}}_{2}S_{s}{+H}_{2}O$$

The whole reaction is:(38)$${4\mathrm{Ni}+2S}^{2-}{+O}_{2}{+2H}_{2}O↔2{\mathrm{Ni}}_{2}{S+4\mathrm{OH}}^{-}$$

The formation of hydroxide proceeds through the following reactions:(39)$${\mathrm{Ni}}^{+}{}_{\mathrm{aq}}{+\mathrm{OH}}^{-}{}_{\mathrm{aq}}\to \mathrm{Ni}{\left(\mathrm{OH}\right)}_{s}$$

Although Ni-sulfide is not found in the corrosion products, the Ni content in the corrosion products significantly decreases with immersion time. This process is verified by the results of the EDS analysis.

## 5. Conclusions

(1) The initial corrosion rate of C71500 copper-nickel alloy in the polluted marine environment is much higher. As the immersion time increases, the corrosion rate greatly decreases, and finally stabilizes in the range of 0.09 mm/a.

(2) The C71500 alloy is in activated corrosion state in the polluted marine environment. Diffusion resistance can be observed at the early stage where a certain protective corrosion product film is formed and tends to stabilize with immersion time, thus reducing the corrosion rate.

(3) With the extension of immersion time, the corrosion product of C71500 alloy begins to clump and adhere to the surface of the sample and becomes dense with sharp edges and corners, and the corrosion products change from spherical to polygonal.

(4) The corrosion product film is mainly Cu_2_S in the early stage of immersion, and Cu_2_O in the later stage; the content of Mn^2+^ is relatively stable; the oxides are mainly Cu_2_O and Ni_2_O_3_; the sulfide is less present in the early immersion stage and gradually increases in the later stage. It is mainly composed of Cu_2_S, CuS, and FeS_2_, and a stable Cu_2_(OH)_3_Cl layer is formed.

(5) The corrosion mechanism of C71500 alloy in sulfide-polluted seawater is similar to that in conventional seawater in the early stages of immersion; however, Cu_2_S gradually replaces the reaction process of Cu_2_O with immersion time, thus accelerating the corrosion. At the same time, with the dissolution reaction of increased S^2−^, Ni, and Fe on the surface of the alloy, the corrosion resistance of the alloy further reduces.

## Figures and Tables

**Figure 1 materials-15-08513-f001:**
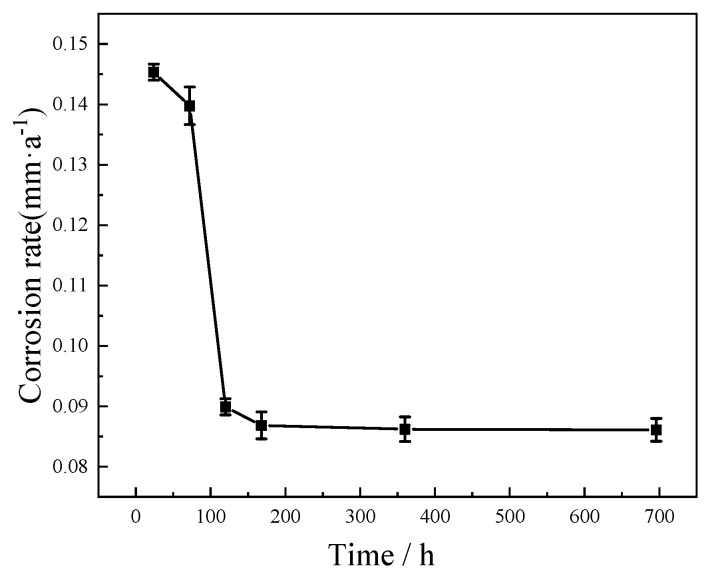
Corrosion rate of C71500 alloy after immersion in simulated polluted marine environment at room temperature for different times (24, 72, 120, 168, 336, and 672 h).

**Figure 2 materials-15-08513-f002:**
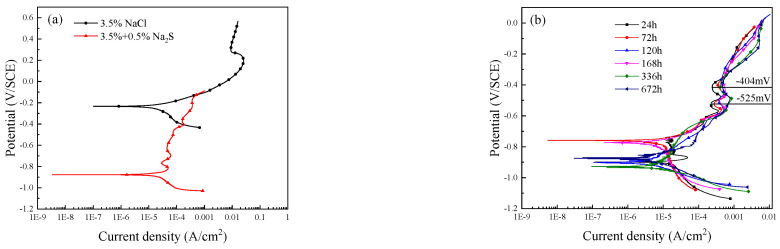
Polarization curves of C71500 alloy under different corrosion conditions: (**a**) immersion in 3.5 wt.% NaCl solution for 30 min with and without sulfide; (**b**) immersion in 3.5 wt.% NaCl solution containing sulfide for different times.

**Figure 3 materials-15-08513-f003:**
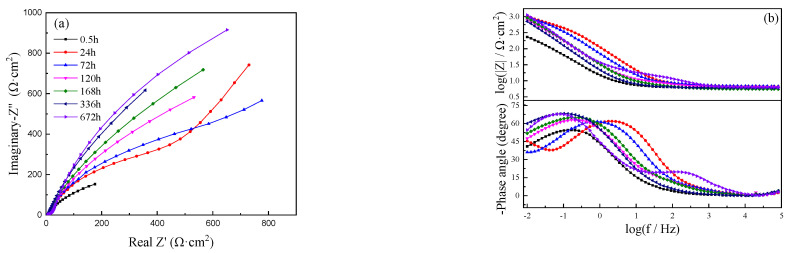
EIS of C71500 alloy in simulated polluted marine solution. (**a**) Nyquist; (**b**) Bode.

**Figure 4 materials-15-08513-f004:**
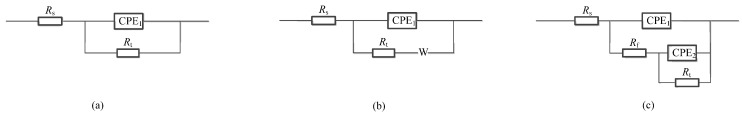
Equivalent circuits of C71500 alloy in simulated polluted marine solution: (**a**) 0.5 h; (**b**) 24 h, 72 h; and (**c**) 120 h.

**Figure 5 materials-15-08513-f005:**
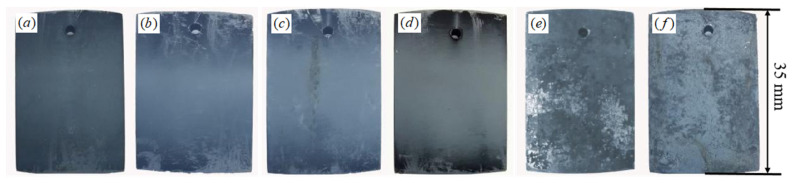
Macroscopic morphology of sample after immersion for different times: (**a**) 24 h; (**b**) 72 h; (**c**) 120 h; (**d**) 168 h; (**e**) 336 h; (**f**) 672 h.

**Figure 6 materials-15-08513-f006:**
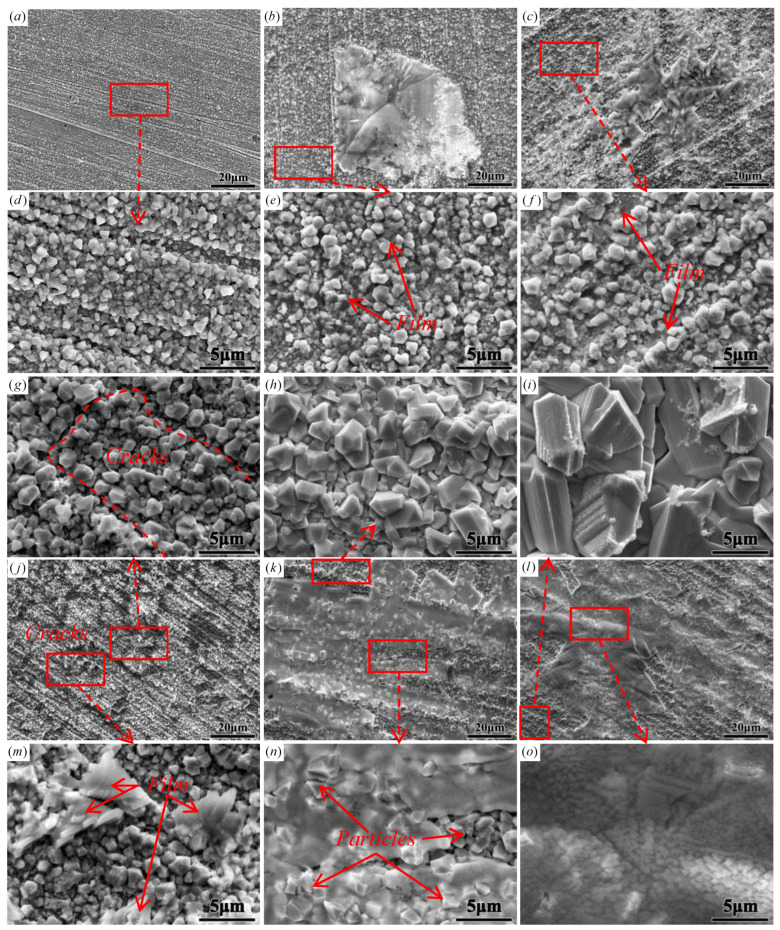
SEM morphology of the corrosion product layer of the alloy after different immersion times: (**a**,**d**) 24 h; (**b**,**e**) 72 h; (**c**,**f**) 120 h; (**g**,**j**,**m**) 168 h; (**h**,**k**,**n**) 336 h; (**i**,**l**,**o**) 672 h. The arrows and red boxes represent the original positions and relationships of the enlarged image.

**Figure 7 materials-15-08513-f007:**
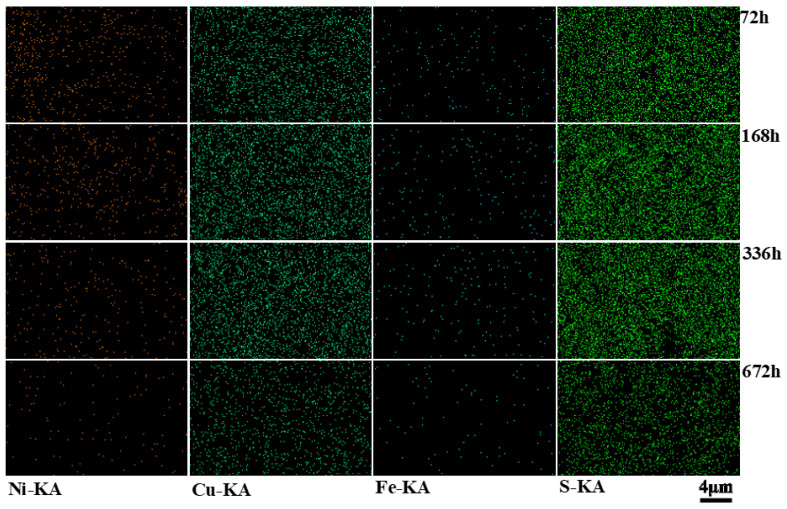
Scanning energy spectrum analysis of C71500 alloy surface corrosion products after different immersion times (72, 168, 336, and 672 h).

**Figure 8 materials-15-08513-f008:**
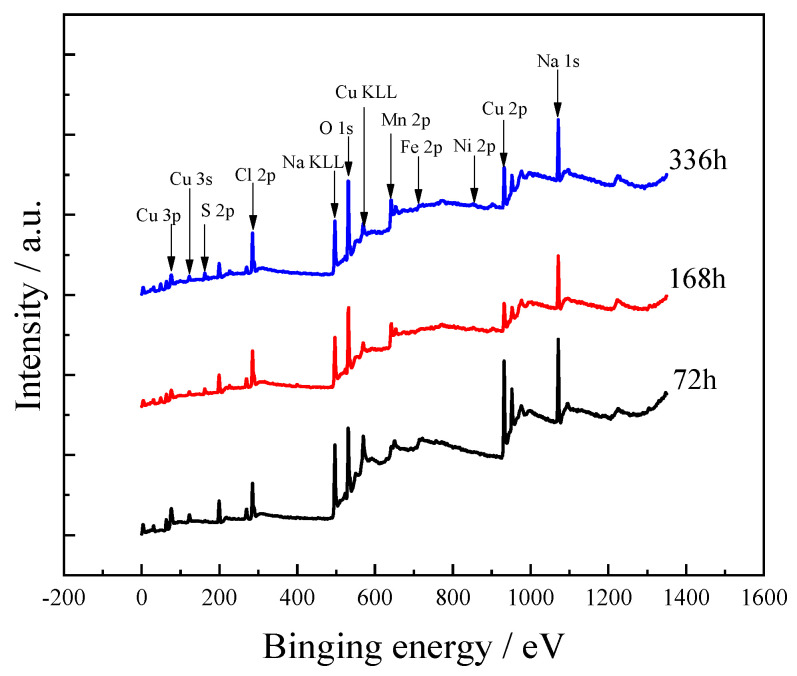
XPS spectrum of C71500 alloy corroded samples after various immersion times (72, 168, and 336 h).

**Figure 9 materials-15-08513-f009:**
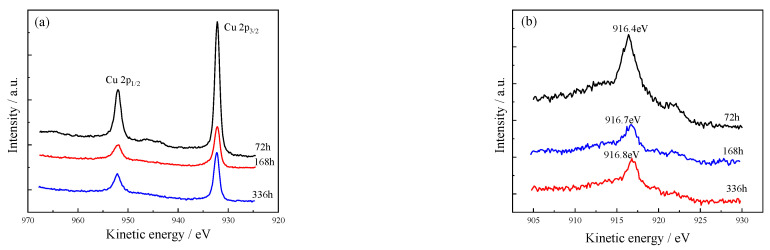
Cu spectrum of alloy after different corrosion times (72, 168, and 336 h). (**a**) Cu 2p; (**b**) Cu CLM.

**Figure 10 materials-15-08513-f010:**
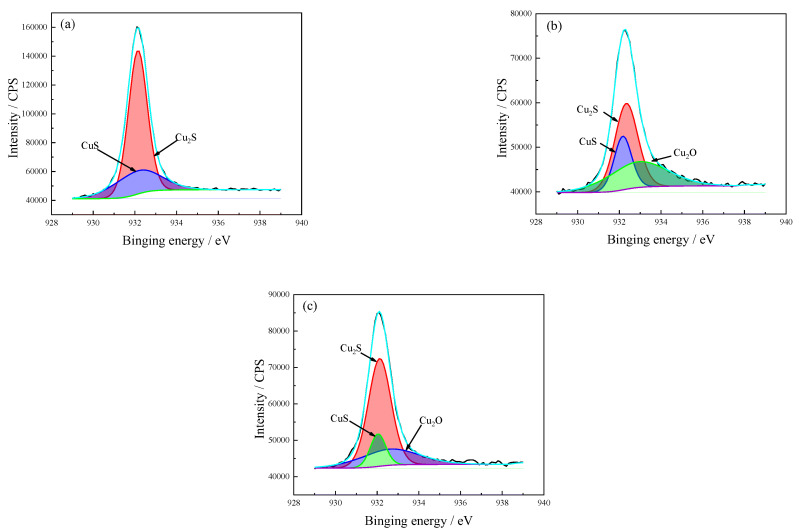
Cu 2p3/2 spectrum of C71500 alloy corrosion product layer after different immersion times. (**a**) 72 h; (**b**) 168 h; (**c**) 336 h. The black line is the original result and the color line is the fitting result.

**Figure 11 materials-15-08513-f011:**
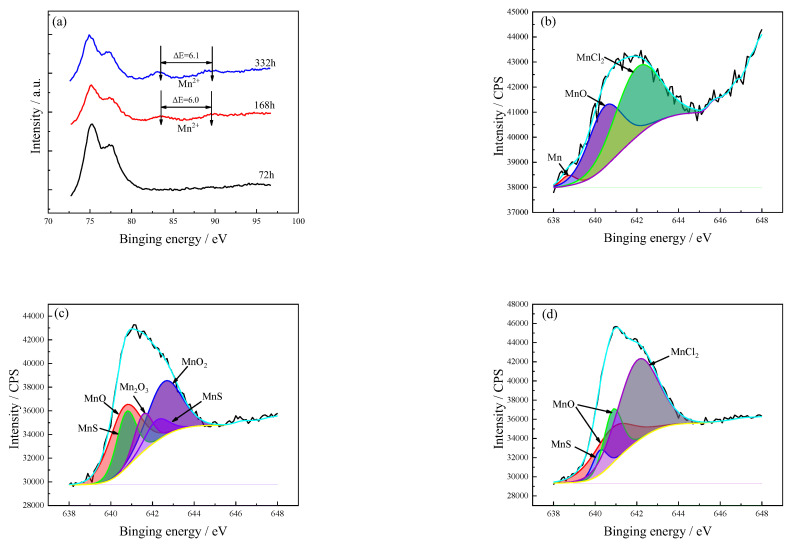
The corrosion product layer of Mn spectra of C71500 alloy after different immersion times. (**a**) Mn 3s; (**b**) 72 h; (**c**) 168 h; (**d**) 336 h. The black line is the original result and the color line is the fitting result.

**Figure 12 materials-15-08513-f012:**
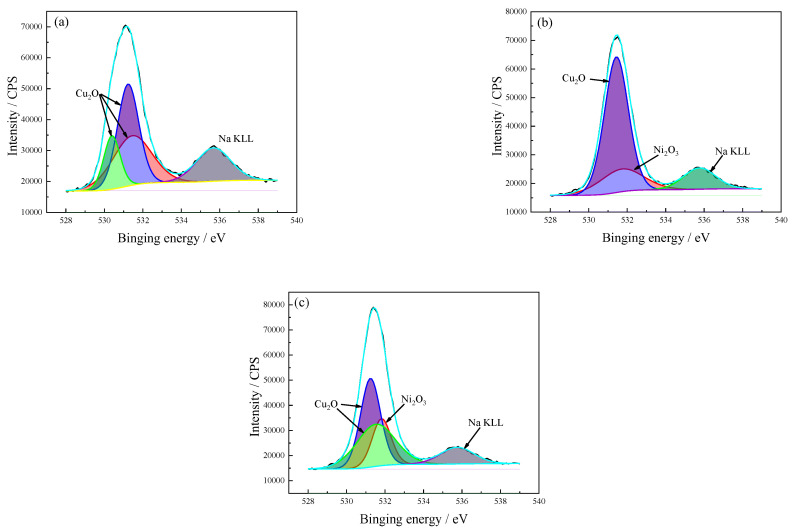
O 1s spectrum of the C71500 alloy surface corrosion product layer after different immersion times. (**a**) 72 h; (**b**) 168 h; (**c**) 336 h. The black line is the original result and the color line is the fitting result.

**Figure 13 materials-15-08513-f013:**
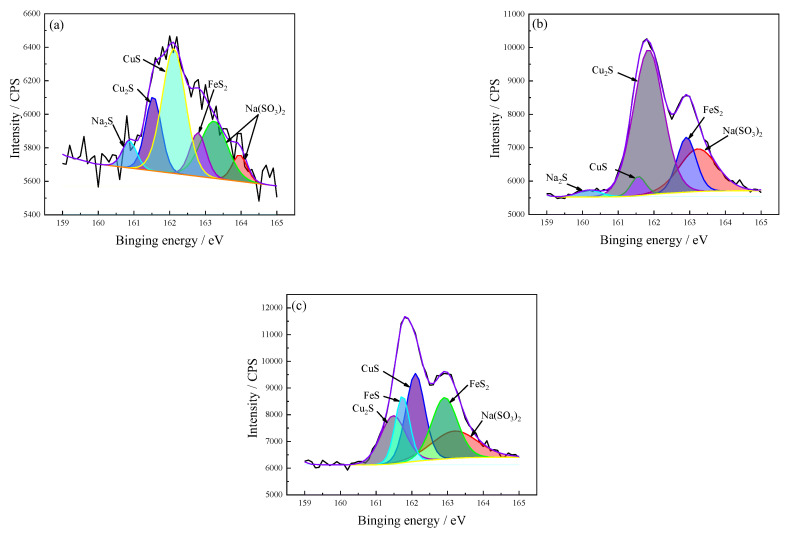
The corrosion product S 2p spectra after different immersion times. (**a**) 72 h; (**b**) 168 h; (**c**) 336 h. The black line is the original result and the color line is the fitting result.

**Figure 14 materials-15-08513-f014:**
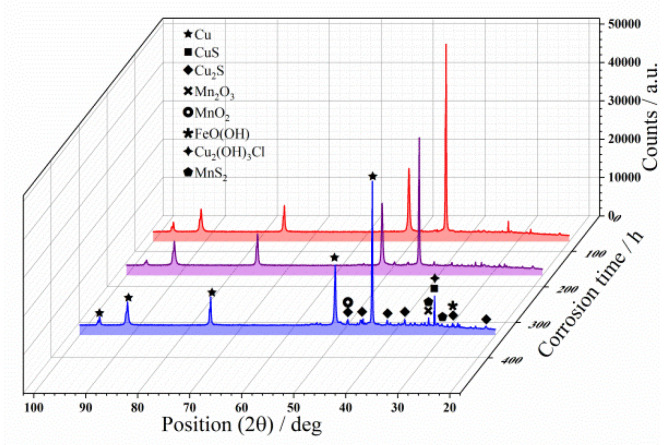
XRD analysis of C71500 alloy surface corrosion product after 72-h, 168-h, and 332-h corrosion.

**Figure 15 materials-15-08513-f015:**
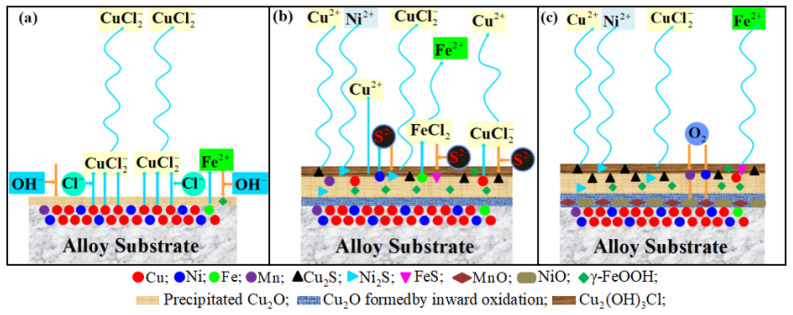
Schematic diagram of the formation process of corrosion product film on C71500 copper-nickel alloy in heavily polluted seawater environment: (**a**) the initial stage of corrosion, copper dissolves to form ${\mathrm{CuCl}}_{2}^{-}$ and Cu_2_O precipitation layer is formed; (**b**) the middle stage of corrosion, unprotected ${\mathrm{Cu}}_{2}{\left(\mathrm{OH}\right)}_{3}\mathrm{Cl}$ is formed, Cu_2_O film starts to grow endogenous, the Cu_2_O precipitation layer thickens, S^2+^ leads to the formation of Cu_2_S, Ni_2_S, and Fe_2_S, and the accelerated dissolution of Fe and Ni; (**c**) the later stage of corrosion, diffused oxygen forms MnO and NiO in the Cu_2_O precipitation layer, and the corrosion performance of the alloy increases.

**Table 1 materials-15-08513-t001:** Characteristic parameters of polarization curve of alloy fitted for different corrosion times. (Residual is the polarization curve fitting residual error).

Time (h)	*β*_a_ (mV/dec)	*β*_c_ (mV/dec)	*i*_corr_ (μA/cm^2^)	*E*_corr_ (mV)
0.5	337 ± 12	−541 ± 16	27.4 ± 10	−877 ± 25
24	15 ± 7	−18 ± 5	2.72 ± 0.5	−899 ± 8
72	33 ± 5	−41 ± 4	1.7 ± 0.3	−758 ± 7
120	54 ± 4	−30 ± 3	1.3 ± 0.2	−902 ± 8
168	40 ± 5	−23 ± 3	1.2 ± 0.2	−773 ± 6
336	42 ± 3	−23 ± 2	0.8 ± 0.1	−929 ± 7
672	39 ± 3	−42 ± 2	0.6 ± 0.1	−873 ± 5

**Table 2 materials-15-08513-t002:** Electrochemical parameters of EIS obtained by equivalent circuit fitting after different immersion times.

Time, h	*R*_s_, Ω·cm^2^	*R*_t_, Ω·cm^2^	Q_dl_		*R*_f_, Ω·cm^2^	Q_f_		*W*
			*n* _1_	Y_0_ Ω^−1^·cm^−2^·s^n^		*n* _2_	Y_0_ Ω^−1^·cm^−2^·s^n^	10^−4^·Ω^−1^·cm^−2^·s^1/2^
0.5	6.11 ± 1.81	1817 ± 20	0.73 ± 0.10	459 ± 7	--	--	--	--
24	6.96 ± 0.55	2213 ± 12	0.82 ± 0.04	485 ± 6	--	--	--	43.4 ± 2.4
72	7.17 ± 0.51	3454 ± 7	0.77 ± 0.04	908 ± 6	--	--	--	76.4 ± 1.9
120	6.36 ± 0.26	5084 ± 8	0.69 ± 0.03	9 ± 2	3069 ± 15	0.89 ± 0.03	2068 ± 18	--
168	5.53 ± 0.21	5134 ± 6	0.72 ± 0.02	7 ± 1	2617 ± 12	0.89 ± 0.03	2752 ± 11	--
336	5.98 ± 0.16	5950 ± 6	0.78 ± 0.01	9 ± 1	2658 ± 8	0.90 ± 0.02	3164 ± 9	--
672	6.17 ± 0.12	6619 ± 4	0.71 ± 0.01	15 ± 1	2686 ± 7	0.84 ± 0.01	3844 ± 10	--

**Table 3 materials-15-08513-t003:** Corrosion products EDS analysis of C71500 alloy after immersion for different times (wt.%).

Exposure Time (h)	O	S	Mn	Fe	Ni	Cu
24	0.4	6.2	0.8	1.7	30.8	60.1
72	2.6	18.1	1.3	0.6	5.5	71.9
120	4.7	13.0	1.1	0.5	6.9	73.9
168	0.8	17.9	0.3	0.3	6.4	74.4
336	1.3	23.5	0.6	0.2	1.0	73.4
672	0.3	18.4	1.6	0.2	0.6	78.8

**Table 4 materials-15-08513-t004:** The corrosion product phase analysis of Cu 2p3/2 spectrum after different immersion times.

Valence	Exposure Time (h)	Proposed Compounds	Binding Energy (eV)	Intensity Area	Atomic (%)
Cu 2p3/2	72	Cu_2_S	932.3	115,144	71.4
		CuS	932.1	46,123	28.6
	168	Cu_2_S	932.3	27,119	44.3
		CuS	932.2	12,969	21.2
		Cu_2_O	932.9	21,124	34.5
	336	Cu_2_S	932.1	41,307	62.6
		CuS	932.0	8467	12.8
		Cu_2_O	932.6	16,175	24.5

**Table 5 materials-15-08513-t005:** Mn 2p spectrum phase analysis of corrosion product after different immersion times.

Valence	Exposure Time (h)	Proposed Compounds	Binding Energy (eV)	Intensity Area	Atomic (%)
Mn 2p3/2	72	Mn	638.7	367	2.8
		MnO	640.5	5061	39.0
		MnO	640.7	10,260	32.4
	168	MnS	640.7	5713	18.1
		MnS	642.2	2526	8.0
		MnO	640.8	9187	23.9
		MnCl_2_	642.0	7557	58.2
		MnO_2_	642.6	9584	30.3
	336	MnS	640.2	2266	5.9
		MnO	640.9	6228	16.2
		MnO	640.8	9187	23.9
		MnCl_2_	642.1	20,745	54.0

**Table 6 materials-15-08513-t006:** Corrosion product phase analysis of O 1s spectrum after different immersion times.

Valence	Exposure Time (h)	Proposed Compounds	Binding Energy (eV)	Intensity Area	Atomic (%)
O 1s	72	Cu_2_O	531.5	41,584	31.5
		Cu_2_O	531.2	47,032	35.6
		Cu_2_O	530.4	18,914	14.3
		Na KLL	535.7	24,612	18.7
	168	Cu_2_O	531.4	75,394	66.1
		Ni_2_O_3_	531.7	22,763	20.0
		Na KLL	535.8	15,830	13.9
	336	Cu_2_O	531.2	46,810	36.5
		Cu_2_O	531.5	43,086	33.6
		Ni_2_O_3_	531.8	23,414	18.3
		Na KLL	535.8	14,872	11.6

**Table 7 materials-15-08513-t007:** The corrosion product phase analysis of S 2p spectrum after different corrosion times.

Valence	Exposure Time (h)	Proposed Compounds	Binding Energy (eV)	Intensity Area	Atomic (%)
S 2p	72	Cu_2_S	161.5	254	16.8
		CuS	162.1	620	40.8
		FeS_2_	162.8	150	9.9
		Na_2_S	160.9	84	5.5
		Na(SO_3_)_2_	163.2	327	21.5
		Na(SO_3_)_2_	164.0	83	5.5
	168	Cu_2_S	161.9	4654	58.4
		CuS	161.6	300	3.8
		FeS_2_	162.9	1132	14.2
		Na_2_S	160.2	174	2.2
		Na(SO_3_)_2_	163.2	1713	21.5
	336	Cu_2_S	161.7	1310	14.6
		CuS	162.1	2226	24.8
		FeS_2_	162.9	2131	23.8
		FeS	161.5	1619	18.1
		Na(SO_3_)_2_	163.2	1673	18.7

## Data Availability

No data were used to support this study.

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
