# Peer review of "Corrosion Behavior of High-Strength C71500 Copper-Nickel Alloy in Simulated Seawater with High Concentration of Sulfide"

_materials, 2022, doi:10.3390/ma15238513_

Round 1

Reviewer 1 Report

The presented manuscript seems to be interesting for readers of the “Materials” journal, it is written in a good manner and suits the requirements of the journal. It can be accepted for publication after minor corrections listed below.

- The captions of the figures are very brief and should be rewritten. Each part of the figure should be explained separately in the caption.

 - The novelty of your work should be presented better at the end of introduction part. This should be presented with more details.

- Abbreviation acronyms, such as AETDA, should all be defined at their first occurrence in the manuscript

- All parameters used in the equations should be described. It is suggested to add a section for the acronyms and parameters at the end of the manuscript.

It is suggested to change the title of section 2 from “2. Experimental” to “2. Experimental procedure”.

- Microscopic structure of C71500 copper-nickel alloy and characteristics of grain size, precipitates, etc. should be given.

- The parameters and units in formula 1 are unclear and should be rewritten.

- Image components in Figure 5 should be brought to a higher magnification and scale.

- The title of the last section should be changed from “4. Conclusions” to “5. Conclusions”.

- In the "Conclusion" section, the authors should present more quantitative data as the main results of the research study rather than just some qualitative data.

- Literature review is not sufficient and authors must review and cite more papers in the field and especially newly published ones. Doing this, review and citing the following refs could be helpful:

[] Metals, 10, 2020, 634.

[] Journal of Materials Engineering and Performance, 27, 2018, 271-281.

Author Response

- The captions of the figures are very brief and should be rewritten. Each part of the figure should be explained separately in the caption.

Response: We have made corresponding modifications.

- The novelty of your work should be presented better at the end of introduction part. This should be presented with more details.

Response: The purpose and significance of this paper has been re-written in the Introduction part.

- Abbreviation acronyms, such as AETDA, should all be defined at their first occurrence in the manuscript

Response: We have modified this error.

- All parameters used in the equations should be described. It is suggested to add a section for the acronyms and parameters at the end of the manuscript.

Response:All the formula parameters have been indicated, so there is no need to add more information at the end of the article

It is suggested to change the title of section 2 from “2. Experimental” to “2. Experimental procedure”.

Response: We have modified this error.

- Microscopic structure of C71500 copper-nickel alloy and characteristics of grain size, precipitates, etc. should be given.

Response: Marks have been added in the corresponding positions in the paper. For the figure of grain boundary size, please refer to our previous published papers [13-17].

- The parameters and units in formula 1 are unclear and should be rewritten.

Response: We have modified this error.

- Image components in Figure 5 should be brought to a higher magnification and scale.

Response: This is a macro photo, we set to the size for typography.

- The title of the last section should be changed from “4. Conclusions” to “5. Conclusions”.

Response: We have modified this error.

- In the "Conclusion" section, the authors should present more quantitative data as the main results of the research study rather than just some qualitative data.

Response: We have made corresponding modifications.

- Literature review is not sufficient and authors must review and cite more papers in the field and especially newly published ones. Doing this, review and citing the following refs could be helpful:

[] Metals, 10, 2020, 634.

[] Journal of Materials Engineering and Performance, 27, 2018, 271-281.

Response: We have added the above mentioned literature.

Reviewer 2 Report

1. The Authors must improve the formatting of the manuscript - the font size in the text is not the same, also some parts are highlighted in yellow etc.

2. For the results from the potentiodynamic polarization tests shown in Table 1 the Authors should add standard deviation values. Potentiodynamic polarization tests are known for a significant scattering of the results. Therefore, adding standard deviations is mandatory in order to prove, that obtained results are not within error limits and that there are some differences between the samples.

3. Results in Table 2 - please add standard deviation values.

4. Figure 5 - there is a lack of a scale bar.

5. Figure 6 - it is not clear how the Authors distinguish particles from layers, especially that layers are also not continuous and there are cracks.

6. The images showing EDS results of corrosion products in Figure 7 must be improved. The quality of the image is poor, maybe a longer time of the measurement is needed to obtained better maps.

Author Response

  1. The Authors must improve the formatting of the manuscript - the font size in the text is not the same, also some parts are highlighted in yellow etc.

Response: We have modified this error.

  1. For the results from the potentiodynamic polarization tests shown in Table 1 the Authors should add standard deviation values. Potentiodynamic polarization tests are known for a significant scattering of the results. Therefore, adding standard deviations is mandatory in order to prove, that obtained results are not within error limits and that there are some differences between the samples.

Response: We have made corresponding modifications.

  1. Results in Table 2 - please add standard deviation values.

Response: We have made corresponding modifications.

  1. Figure 5 - there is a lack of a scale bar.

Response: We have added the scale bar.

  1. Figure 6 - it is not clear how the Authors distinguish particles from layers, especially that layers are also not continuous and there are cracks.

Response: The grain is a single crystal with polygonal structure, and the lamellar is a continuous interface without obvious crystal structure.

  1. The images showing EDS results of corrosion products in Figure 7 must be improved. The quality of the image is poor, maybe a longer time of the measurement is needed to obtained better maps.

Response: The images in Figure is mainly intended to describe the chemical distribution of corrosion products, so the selected field of view is relatively large, which could be representative.

Reviewer 3 Report

Reviewer Recommendation and Comments for manuscript materials-2029221 with the title: “Corrosion behavior of high- strength C71500 copper-nickel alloy in simulated seawater with high concentration of sulfide”, authors: X. Gao, M. Liu.

The authors present the corrosion behavior of C71500 alloy in seawater studied by potentiodynamic polarization, electrochemical impedance spectroscopy, immersion tests, and combined with SEM, EDS, XPS, XRD surface analysis methods..

The main comments that I find useful for improving the quality of the article are presented below:

*The manuscript should be checked and proofread by a native English speaker.

*L107. “where: v, yr, m0, g, m1, S0, m2, t, h and d is…”. All this is not found in Formula 1. (d is kg/m3 ?)

*L136. Figure 1 – Y-axis “(mm·a-1)”. “a” is year? (mm/y ?)

*Figure 1 must be improved. Is unclear.

*L100. “6 durations of 24, 72, 120, 168, 336, and 672 h.”. L138. “for different times (24, 72, 120, 168, 336, and 672 h).”. The points shown in Figure 1 do not correspond to these times. For example, there are determinations at 530 minutes!

*Corrosion rate of C71500 alloy after immersion in 3.5 wt.% NaCl + 0.5 wt.% Na2S solution for 24 h must be 0.006 mm/y. (please see Int. J. Electrochem. Sci., 16 (2021) Article ID: 210224, doi: 10.20964/2021.02.35).  Why are there such big differences for the same alloy under the same conditions?

*L141. “0.15 mm/a”. “a” is year? (mm/y ?)

*L142. “When the corrosion time reaches 168 h,” Stabilization can be observed as early as 120 mins!?

*L143. “0.09 mm/yr.” ? (mm/y ?)

*Please check and recheck the icorr values in Figure 2 and those shown in Table 1. Figure 2b indicates values of icorr of about IE-5 A/cm2 (10 mA/cm2)!?

*0.5h+20C _ icorr = 96.6 (this manuscript)

 24h+20C _ icorr = 3.2 (this manuscript)

0.5h+20C _ icorr = 27.4 (Int. J. Electrochem. Sci., 16 (2021) ID: 210224, doi: 10.20964/2021.02.35)

24h+20C _ icorr = 2.72 (Int. J. Electrochem. Sci., 16 (2021) ID: 210224, doi: 10.20964/2021.02.35)

Why are there such big differences for the same alloy under the same conditions?

*EIS results for Nyquist and Bode diagrams do not correspond at all!?. Partial results are published and do not are the same!

*EIS equivalent circuit is not the same in the two articles!? For the same alloy in the same conditions!?

*Partial Figure 6 is already published!?

*Mechanism is partial published!?

*There are figures already published!

*A figure cannot be published more than once!

*The typos must be corrected.

L65. technology,,

L67. environment[17].

L115. “other electrochemical testing was measured

etc.

*The Materials journal require a specific format of references, authors must pay more attention in their writing. No reference is written according to the format required by the journal.

*There are some grammar and typing mistakes.

*The authors must revise the entire manuscript.

Author Response

*The manuscript should be checked and proofread by a native English speaker.

Response: We have asked the native English speaker to check the whole manuscript.

*L107. “where: v, yr, m0, g, m1, S0, m2, t, h and d is…”. All this is not found in Formula 1. (d is kg/m3 ?)

Response: We have rewritten this part.

*L136. Figure 1 – Y-axis “(mm·a-1)”. “a” is year? (mm/y ?)

Response: Correct.

*Figure 1 must be improved. Is unclear.

Response: We have provided the Origin image.

*L100. “6 durations of 24, 72, 120, 168, 336, and 672 h.”. L138. “for different times (24, 72, 120, 168, 336, and 672 h).”. The points shown in Figure 1 do not correspond to these times. For example, there are determinations at 530 minutes!

Response: In Figure 1, a set of 528 h (22 days) data was added to observe data continuity.

*Corrosion rate of C71500 alloy after immersion in 3.5 wt.% NaCl + 0.5 wt.% Na2S solution for 24 h must be 0.006 mm/y. (please see Int. J. Electrochem. Sci., 16 (2021) Article ID: 210224, doi: 10.20964/2021.02.35).  Why are there such big differences for the same alloy under the same conditions?

Response: This article is the research result of our team, describing the effect of temperature on the corrosion behavior of C71500 copper alloy.

*L141. “0.15 mm/a”. “a” is year? (mm/y ?)

Response: We have changed all “mm/y” to “mm/a”.

*L142. “When the corrosion time reaches 168 h,” Stabilization can be observed as early as 120 mins!?

Response: After 168 h immersion, the corrosion rate is more stable.

*L143. “0.09 mm/yr.” ? (mm/y ?)

Response: We have changed all “mm/yr.” to “mm/a”.

*Please check and recheck the icorr values in Figure 2 and those shown in Table 1. Figure 2b indicates values of icorr of about IE-5 A/cm2 (10 mA/cm2)!?

*0.5h+20C _ icorr = 96.6 (this manuscript)

 24h+20C _ icorr = 3.2 (this manuscript)

0.5h+20C _ icorr = 27.4 (Int. J. Electrochem. Sci., 16 (2021) ID: 210224, doi: 10.20964/2021.02.35)

24h+20C _ icorr = 2.72 (Int. J. Electrochem. Sci., 16 (2021) ID: 210224, doi: 10.20964/2021.02.35)

Why are there such big differences for the same alloy under the same conditions?

Response: The surface finish of the sample, surface oxidation state has a great impact on the corrosion rate.

*EIS results for Nyquist and Bode diagrams do not correspond at all!?. Partial results are published and do not are the same!

*EIS equivalent circuit is not the same in the two articles!? For the same alloy in the same conditions!?

Response: The microstructure and trace element control of the alloy are different.

*Partial Figure 6 is already published!?

Response: The Fig. 6(a) has been used, and the two has been changed.

*Mechanism is partial published!?

*There are figures already published!

*A figure cannot be published more than once!

*The typos must be corrected.

L65. technology,,

L67. environment[17].

L115. “other electrochemical testing was measured”

etc.

Response: We have modified those errors.

*The Materials journal require a specific format of references, authors must pay more attention in their writing. No reference is written according to the format required by the journal.

Response: We have modified this error.

*There are some grammar and typing mistakes.

Response: We have made corresponding modifications.

*The authors must revise the entire manuscript.

Response: We have made corresponding modifications.

Round 2

Reviewer 1 Report

As authors have performed an adequate revise, the manuscript might be accepted for publication in the journal of Materials.

Author Response

Thanks very much for your kind advice.

Reviewer 2 Report

Authors answered to all my comments.

Author Response

Thanks very much for your kind advice.

Reviewer 3 Report

*L100. “6 durations of 24, 72, 120, 168, 336, and 672 h.”. L138. “for different times (24, 72, 120, 168, 336, and 672 h).”. The points shown in Figure 1 do not correspond to these times. For example, there are determinations at 530 minutes!

Response: In Figure 1, a set of 528 h (22 days) data was added to observe data continuity.

*The data shown in the figure are not consistent with the experimental procedure.

*Corrosion rate of C71500 alloy after immersion in 3.5 wt.% NaCl + 0.5 wt.% Na2S solution for 24 h must be 0.006 mm/y. (please see Int. J. Electrochem. Sci., 16 (2021) Article ID: 210224, doi: 10.20964/2021.02.35). Why are there such big differences for the same alloy under the same conditions?

Response: This article is the research result of our team, describing the effect of temperature on the corrosion behavior of C71500 copper alloy.

*Why are there such big differences for the same alloy under the same conditions? Authors' published data must show reproducibility / for the same material and the same corrosive environment.

*L142. “When the corrosion time reaches 168 h,” Stabilization can be observed as early as 120 mins!?

Response: After 168 h immersion, the corrosion rate is more stable.

*What is the method by which 'more' is assessed?

*Please check and recheck the icorr values in Figure 2 and those shown in Table 1. Figure 2b indicates values of icorr of about IE-5 A/cm2 (10 mA/cm2)!?

*0.5h+20C _ icorr = 96.6 (this manuscript)

24h+20C _ icorr = 3.2 (this manuscript)

0.5h+20C _ icorr = 27.4 (Int. J. Electrochem. Sci., 16 (2021) ID: 210224, doi: 10.20964/2021.02.35)

24h+20C _ icorr = 2.72 (Int. J. Electrochem. Sci., 16 (2021) ID: 210224, doi: 10.20964/2021.02.35)

Why are there such big differences for the same alloy under the same conditions?

Response: The surface finish of the sample, surface oxidation state has a great impact on the corrosion rate.

*Authors' published data must show reproducibility / for the same material and the same corrosive environment. The tests must be repeated with the certainty that the surface is well finished and that there are no oxidation compounds on the surface.

*EIS results for Nyquist and Bode diagrams do not correspond at all!?. Partial results are published and do not are the same!

*No answer.

*EIS equivalent circuit is not the same in the two articles!? For the same alloy in the same conditions!?

Response: The microstructure and trace element control of the alloy are different.

*The alloy is the same, C71500! Or not?

*Partial Figure 6 is already published!?

Response: The Fig. 6(a) has been used, and the two has been changed.

*The surface analysis carried out last year cannot be published and the electrochemical analysis done this year for the same alloy with different structures?

Figure 6(d) was published in Int. J. Electrochem. Sci., 16 (2021) ID: 210224, doi: 10.20964/2021.02.35

*Mechanism is partial published!?

*No answer.

*There are figures already published!

*No answer.

*A figure cannot be published more than once!

*No answer.

Author Response

*L100. “6 durations of 24, 72, 120, 168, 336, and 672 h.”. L138. “for different times (24, 72, 120, 168, 336, and 672 h).”. The points shown in Figure 1 do not correspond to these times. For example, there are determinations at 530 minutes!

Response: In Figure 1, a set of 528 h (22 days) of data was added to observe data continuity.

*The data shown in the figure are not consistent with the experimental procedure.

 Response: We have deleted the data of 528 h.

*Corrosion rate of C71500 alloy after immersion in 3.5 wt.% NaCl + 0.5 wt.% Na2S solution for 24 h must be 0.006 mm/y. (please see Int. J. Electrochem. Sci., 16 (2021) Article ID: 210224, doi: 10.20964/2021.02.35). Why are there such big differences for the same alloy under the same conditions?

Response: This article is the research result of our team, describing the effect of temperature on the corrosion behavior of C71500 copper alloy.

*Why are there such big differences for the same alloy under the same conditions? Authors' published data must show reproducibility / for the same material and the same corrosive environment.

Response: There are differences between the two test environments. The previous test was conducted in an open fume hood, while the present test was conducted in a sealed container. The pH of this experiment was regulated by dilute hydrochloric acid, and the reaction between Na2S and dilute hydrochloric acid would produce volatilized hydrogen sulfide. Relatively speaking, the pH value of the corrosion solution measured by the sealed container in this experiment is more stable and the experimental results are more accurate, so the results are different. In particular, our original test data (including the results of different pH values) are attached, so this data cannot be adjusted.

No.

outer diameter

length

width

 thickness

area

m1

m2

m1-m2

time

Corrosion rate

T

pH

1

61.2

33.84

23.11

4.99

2053.12

32.6996

32.6923

0.0073

24

0.145818

20

8.2

2

61.2

33.69

21.44

5.03

1921.14

29.8745

29.8538

0.0207

72

0.147297

20

8.2

3

61.2

33.61

22.97

5.03

2033.58

32.3236

32.3013

0.0223

120

0.089945

20

8.2

4

61.2

34.03

23.17

4.96

2065.3

32.6839

32.6532

0.0307

168

0.087088

20

8.2

5

61.2

33.59

23.01

5.02

2034.7

32.2962

32.2362

0.06

336

0.086382

20

8.2

6

61.2

33.57

22.95

4.99

2026.22

31.8065

31.7424

0.0641

360

0.086493

20

8.2

7

61.2

33.58

23.13

5.02

2043.35

32.476

32.3811

0.0949

528

0.086577

20

8.2

8

61.2

33.36

22.26

4.96

1960.37

30.9163

30.7962

0.1201

696

0.086638

20

8.2

9

61.2

34.04

22.97

4.95

2049.08

32.4814

32.4148

0.0666

24

1.332963

20

2

10

61.2

33.99

21.64

4.99

1948.05

30.1938

30.1507

0.0431

24

0.907361

20

4

11

61.2

33.65

21.51

5.05

1926.16

30.0484

30.0055

0.0429

24

0.913414

20

6

12

61.2

33.84

23.11

4.99

2053.12

32.6996

32.6923

0.0073

24

0.145818

20

8.2

13

61.2

33.53

23.23

5.02

2048.32

32.435

32.4325

0.0025

24

0.050055

20

10

14

61.2

34.61

22.28

5.01

2031.07

32.0416

32.0415

0.0001

24

0.002019

20

11.79

*L142. “When the corrosion time reaches 168 h,” Stabilization can be observed as early as 120 mins!?

Response: After 168 h immersion, the corrosion rate is more stable.

*What is the method by which 'more' is assessed?

 Response: The more stable way to evaluate is that the weight loss does not change with the increase of time, and we have modified it according to your suggestion.

*Please check and recheck the icorr values in Figure 2 and those shown in Table 1. Figure 2b indicates values of icorr of about IE-5 A/cm2 (10 mA/cm2)!?

*0.5h+20C _ icorr = 96.6 (this manuscript)

24h+20C _ icorr = 3.2 (this manuscript)

0.5h+20C _ icorr = 27.4 (Int. J. Electrochem. Sci., 16 (2021) ID: 210224, doi: 10.20964/2021.02.35)

24h+20C _ icorr = 2.72 (Int. J. Electrochem. Sci., 16 (2021) ID: 210224, doi: 10.20964/2021.02.35)

Why are there such big differences for the same alloy under the same conditions?

Response: The surface finish of the sample, and surface oxidation state have a great impact on the corrosion rate.

*Authors' published data must show reproducibility / for the same material and the same corrosive environment. The tests must be repeated with the certainty that the surface is well-finished and that there are no oxidation compounds on the surface.

Response: We have modified those errors.

*EIS results for Nyquist and Bode diagrams do not correspond at all!?. Partial results are published and do not are the same!

Response: The test environments are different, we have answered in the weight loss test part.

*EIS equivalent circuit is not the same in the two articles!? For the same alloy in the same conditions!?

Response: The microstructure and trace element control of the alloy is different.

*The alloy is the same, C71500! Or not?

Response: The test environments are different, we have answered in the weight loss test part.

*Partial Figure 6 is already published!?

Response: Fig. 6(a) has been used, and the two have been changed.

*The surface analysis carried out last year cannot be published and the electrochemical analysis did this year for the same alloy with different structures?

Figure 6(d) was published in Int. J. Electrochem. Sci., 16 (2021) ID: 210224, doi: 10.20964/2021.02.35

 Response: Fig. 6a and 6d have been changed.

*Mechanism is partially published!?

  Response: The mechanism of the reaction is partly the same, but there are also significant differences, as shown in the schematic diagram.

*There are figures already published!

 Response: The figures have been changed.